# Cross- analyzing the opinions and experiences of nurses, physiotherapists, dentists, midwives, and pharmacists with respect to addictive disorder screening in primary care: A qualitative study

**Agathe Edeline[1], Amelie Tripault[1], Jean Pierre Lebeau[1,2], Maxime Pautrat** [1,2] *

**1** Department of General Practice, Tours Regional University Hospital, Tours, France, **2** Faculty of Medicine, University of Tours, EA7505 Education Ethique Santé, 37000, Tours, France

* maxime.pautrat@univ-tours.fr

## Abstract

Early addiction disorders screening is recommended in primary care. The goal of health system reform is to include allied health professionals in this screening. The appropriation of their new role has not yet been explored. The main aim of this study was to examine the perspective of allied health professionals in primary care on the screening of addictive disorders. This qualitative study inspired by the grounded theory was carried out between August 2018 and July 2019. Semi-structured individual interviews and focus groups were organized to include of primary care health professionals (physiotherapist, nurse, midwife, pharmacist, and dentist). Thirteen semi-structured individual interviews and four focus groups were recorded and coded. The paramedics described the advantages of their professions for the detection of addictions: home visits, prescription history, habit of intimate subjects, close consultations, etc. Despite daily practice-specific observation posts, they sometimes remained silent witnesses, and their helplessness hindered identification. They felt both closer to the patients and less legitimate than the doctors in dealing with addictions. Finally, their desire for a multidisciplinary approach was limited by the fear of disturbing the doctor and the confusion between betrayal and medical secrecy. Paramedical professionals claimed to have a complementary role to play in identifying addictions. Their reluctance echoed the concept of self-censorship, already described in studies with addictologists and patients. These results must be compared with the opinions of general practitioners and patients.

## Introduction

The global burden of addiction disorders is based on their morbidity, mortality, and social costs [1,2]. Alcohol, opioids, and cannabis are the most prevalent, with a risk factor for premature death and disability [3–5]. Non-substance-use addictions, such as gambling, share

**Funding:** The author(s) received no specific funding for this work.

**Competing interests:** The authors have declared that no competing interests exist.

neurobiological and genetic similarities with substance use disorders and have a high rate of comorbidity [6]. The fifth edition of the DSM described eleven criteria for diagnosing dependence, for both drug-related disorders and non-drug-related behaviours [7]. These criteria comes together four: loss of control, physical dependence, social problems, and risky consumption.

Early screening of patients with an addictive disorder reduces their morbidity and mortality and improves their quality of life [8–10]. The Screening Brief Intervention and Referral Treatment (SBIRT) is a prevention tool for professionals, to identify risky substance use among the patients, to reduce it. It has been recommended since 2008, but remains little used by primary care physicians [11,12]. Some addiction screening tests have been validated in primary care [13]. However many already known obstacles to addictive disorder screening in primary care remain, such as lack of time, a feeling of inefficiency, and patient reluctance [14–19].

The World Health Organization (WHO) report of 2018 prompted the implementation of a new policy to prevent addictive disorders [3]. Also in 2018, the French healthcare system reform aimed to systematize and strengthen the screening of addictive disorders, by primary care professionals [20]. General practitioners are no longer the only screeners [20]. Since 2016, dental surgeons, nurses, midwives, and physiotherapists, have been called upon to carry out this screening [21]. In this way, they can prescribe nicotine substitution treatments [20]. Pharmacists are now involved through medication reconciliation [22], such as chronic treatment in addictology. In France, in 2019, advanced practice nurses and medical assistant positions were created to improve the management of chronic conditions, such as addictive disorders [23,24]. The appropriation of this new role of nurses, pharmacists, physiotherapists, midwives, and dentists, has been not explored yet.

The aim of this study was to explore the point of view of primary care paramedics concerning the screening of addictive disorders.

We want to understand the advantages and disadvantages of each primary care profession, in dealing with addictive disorders. Understanding what healthcare professionals think about the screening of addictive disorders, in their day-to-day practice, could help to identify some unknown barriers in their appropriation of the SBIRT protocol. As a first step, it would be useful to explore their ability to screen their patients, and possibly, develop a relevant intervention to encourage them to discuss addiction with their patients and improve early screening.

## Methods

This qualitative study recruited healthcare professionals between August 2018 and July 2019. Using a grounded theory approach, enabled investigators to build a model of healthcare professional's perspectives, on addictive disorder screening in primary care.

The study was carried out in accordance with the Declaration of Helsinki, and approved by the ethic committee""Espace de Réflexion Éthique de la Région Centre", Tours, France (approval number: 2017 059). It is also registered with the Commission nationale de l'informatique et des libertés. This research exploring the practices and views of healthcare professionals, did not require authorization from the Institutional Review Board. Each participant signed an informed consent form stating the goals, and reasons for conducting the research. The audio records were destroyed after transcription.

## Participants

Healthcare professionals included physiotherapists, nurses, midwives, pharmacists, and dentists. They were recruited from primary care practices in the Centre Val de Loire, Normandy,

and Ile-de France Regions, France. The first professionals were contacted by phone from the author's caregiver networks, then, others were recruited using a snowball technique.

We conducted eleven individual semi-structured interviews with healthcare professionals, either by telephone or at their place of practice (S1 Appendix). There were seven women and six men, aged between twenty-six and sixty We also conducted multidisciplinary interviews via focus groups including midwives, nurses, and physiotherapists. There were nineteen women and six men, aged between twenty-five and fifty-seven. Great variability in terms of sex, age, method, and practice characteristics, was sought for each professional.

All participants were informed about the study and its objectives and provided informed consent. Only four midwives refused to participate because they did not feel concerned by substance use disorders, in their practice.

## Data collection

Focus groups have the advantage of encouraging interaction between participants, and stimulating inter- and intra-disciplinary exchanges. This method exposes studies to the usual opinion leader and social desirability biases. To limit this desirability bias, along with any external biases, we conducted the focus group in a convivial atmosphere, around lunch. Individual interviews have the advantage of guaranteeing intimacy, spontaneity, and freedom of response during exchanges, on a subject charged with representations, such as addiction.

The initial guild interview was developed by all the authors, and tested on two volunteer caregivers. It included an icebreaker question, which was 'Tell me the story of the last patient with an addiction problem you saw?', an invitation to share experiences of successful and unsuccessful patient screening and their role in the identification process. New reminders were added to explore the concepts emerging from the initial analyses. All interviews and focus groups were audio recorded and transcribed. All verbatim was coded to anonymize participant identity using ph, E1 for pharmacists's interview, for example, FG1 for the first focus group. A personal logbook collected field notes during the research. At the end of the research, all participants were invited to a presentation of verbatims and results, and some of them attended.

## Analysis

The analysis prism was based on the grounded theory approach, which is a research method concerned with the generation of theory through the collecting, and analysis of data. A coding tree was built from many citations of verbatims. These codes were organized in conceptual categories. Finally, a conceptualization was drawn up, based on schematic representations available in the literature. We used the NVivo 11$^{®}$ QSR software for verbatim coding. The scientific validity criteria of the grounded theory analysis were met and thirty two out of thirty two items in the COREQ grid were completed (S2 Appendix), such as data triangulation and inductive analysis [25].

## Results

Thirteen individual interviews and four focus groups were conducted. Data sufficiency was achieved from the eleventh interview, and the third focus group. The characteristics of the participants are detailed in Tables 1 and 2.

## An addiction observation post specific to each professional

Nurses and midwives described the advantages of home visits, where it was possible to observe people's privacy: "*Yeah, because you get the smell, the bottles, even though they try to hide*

**Table 1. Characteristics of the individual interview population.**

|  | Profession, gender and age | Place of practice | Type of work | Interview time | Collected by |
|---|---|---|---|---|---|
| E1 | Ph, Women 28 (ph,E1) | 85 | Rural, ambulatory | 13 min | AT |
| E2 | Ph, Women 37 (ph,E2) | 85 | Rural, ambulatory | 16 min | AT |
| E3 | N, Men 60 (n,E3) | 37 | Rural, ambulatory | 31 min | AE |
| E4 | MW, Women 31 (mw,E4) | 18 | Semi-rural, ambulatory | 22 min | AT |
| E5 | Ph, Men 51 (ph,E5) | 27 | Rural, ambulatory | 20 min | AE |
| E6 | MW, Women 57 (mw,E6) | 37 | Urban, employee in PMI | 48 min | AE |
| E7 | D, Women 35 (d,E7) | 37 | Semi-urban, ambulatory | 21 min | AT |
| E8 | D, Men 51 (d,E8) | 45 | Semi-rural, ambulatory | 21 min | AT |
| E9 | Ph, Men 28 (ph,E9) | 76 | Urban, ambulatory | 43 min | AE |
| E10 | Pt, Men 41 (pt, E10) | 37 | Urban, employee and academic | 49 min | AT |
| E11 | D, Men 59 (d,E11) | 18 | Rural, ambulatory and academic | 20 min | AE |
| E12 | N, Women 49 (n,E12) | 37 | Semi-rural | 47 min | AT |
| E13 | Pt, Women 26 (pt,E13) | 75 | Urban | 23 min | AE |

N : Nurse, MW : MidWife, D : Dentist, Ph : Pharmacist, Pt : Physiotherapist.

AT : Amélie Tripault ; AE : Agathe Edeline.

everything." (nE3) and "*I find that people confide more at home, they welcome you into their homes, we're in their homes, uh, we're around the table, even on their couch*" (mwE6). The pharmacy sometimes became an observatory in the heart of the village: "*Through the window, you*

**Table 2. Characteristics of the focus group population.**

|  | Gender | Place of practice | Type of work | Interview time | Collected by |
|---|---|---|---|---|---|
| FG1 | N, Women 42 (n1,FG1)<br>N, Women 43 (n2,FG1)<br>N, Women 53 (n3,FG1)<br>N, Women 54 (n5,FG1) | 85 | Rural, ambulatory | 53 min | AT |
| FG2 | Pt, Men, 45 (pt1,FG2)<br>Pt, Women, 32 (pt2,FG2)<br>N, Men 49 (n1,FG2)<br>N, Men 52 (n2,FG2)<br>N, Women, 50 (n3,FG2)<br>N, Women 55 (n4,FG2)<br>N, Women 53 (n5,FG2) | 37 | Rural, ambulatory | 50 min | AE |
| FG3 | MW, Women 29 (mw1,FG3)<br>MW, Women 29 (mw2,FG3)<br>MW, Women 41 (mw3,FG3)<br>MW, Women 47 (mw4,FG3)<br>Pt, Women 39 (pt1,FG3)<br>Pt, Men 35 (pt2,FG3)<br>N, Women 57 (n1,FG3) | 37 | Urban, ambulatory | 93 min | AT |
| FG4 | Pt, Men 25 (pt1,FG4)<br>Pt, Men 26 (pt2,FG4)<br>Pt, Women 29 (pt3,FG4)<br>N, Women 37 (n1,FG4)<br>N, Women 42 (n2,FG4)<br>N, Women 46 (n3,FG4)<br>N, Women, 53 (n4,FG4) | 37 | Urban, ambulatory | 43 min | AE |

N : Nurse, MW : MidWife, Pt : Physiotherapist.

AT : Amélie Tripault ; AE : Agathe Edeline.

*can see them anyway, eh, the patients. It overlooks the street and opposite there is a small bar, so you can see what they do during the day*" (phE9). Pharmacists also observed the evolution of addictive disorders thanks to the history of prescriptions: "*Drug addictions are necessarily easier to detect since we have the history*" (phE9). Dentists observed the signs of substance use through the patients' oral condition: "*Well, I can see the mouth, they have a lot of nicotine in their mouths. You can smell it on their breath (laughs)*" (dE8). The physiotherapists noted the no-shows of patients with problematic consumption: "*It's crazy because, as he didn't arrive, I went to get my bread and I met him across the street, at the PMU. . .*" (ptFG4).

## Caregivers' reticence about screening

**Asking the question is like dropping a bomb.** The nurses were reluctant to talk about addictions for fear of the patient's reaction: "*Afterwards, it's true that sometimes, it feels like you're throwing a bomb and then, you're a little afraid of what it's going to bring out in you*" (nFG2), or "*And then, there are reactions that are quite violent (. . .) I've had, What do you care?, I need a blood test, you come in to treat my leg, the rest is none of your business*"(nFG2).

**No questions if no solutions.** The perceived lack of a solution when faced with admitted addiction, led professionals to not ask the question: "*Maybe we're not all very comfortable talking about this subject, because we don't know what to do with it afterward*" (ptE10) and "*I had asked him to talk to more people about it. Because, what are we supposed to do about it? Not much, you know! You have no treatment, you have nothing! And, uh, it's tough*" (phE9).

## Do I go for it, or don't I?

**It's not my place.** Pharmacists didn't always feel legitimate in dealing with addictions, because patients "*also see them as a bit of a salesman so that's the tricky part*" (phE9). Likewise, for dentists: "*Dentists taking care of addictions. . . It's not yet becoming the norm! In people's eyes, we're still technicians!*" (dE11).

The interviews revealed an approach to care that is sometimes organo-centric or prescription-centric. The nurses said "*We come for the care and in general, it suits them very, very well, that we just come for the care*" (nFG4), "*If we ask, we're out of the loop. Outside the act for which we came to see them*" (nFG2) and "*We don't come for the addiction, we come for the bandage, for the injection, for the antibiotics*" (nFG4). The physiotherapists admitted, "*I don't necessarily bring up the subject if it doesn't have an impact on what I do*" (ptFG3). When it occurred, the approach to addictions was limited to the usual technical field of the career: "*I do more smoking prevention because it impacts the gums and the headaches, it affects organs that I treat (laughs)*" (dE7). Faced with a rise in dosage, one pharmacist confided to us that she was not going against the medical prescription: "*It's hard to know where we stop, well, when do we refuse to dispense prescribed drugs?*" (phE1).

When the patient spontaneously brought up the subject of addictions, dentists said that it was not their role: "*I don't really know what makes them tell me*" (dE11) and "*I don't mind if they tell me about it, after all, it's not really my field*" (dE7). They showed a certain disinterest: "*I don't remember addictions, because in my opinion, uh, you come across them without worrying about them*" (dE11).

The healthcare professionals blamed each other, the subject being more and more the colleague's business: "*It's not necessarily up to me to broach the subject, he has a GP, it's not necessarily up to the physiotherapist to be the first, it's not me who gives him primary care for this kind of problem*" (ptFG3), or "*If I don't see him again for a year, uh, it's complicated. It's not like a general practitioner or a physiotherapist, who sees their patient more regularly. . .*" (dE7).

The anchoring of the doctor's status was strong in their minds: "*I think authority is with the doctor*!" (phE9). The doctor seemed to be more legitimate than paramedics, in dealing with the subject of addictions: "*If the doctor asks, it's not indiscretion*" (nFG2) and "*The patient will accept more that the doctor asks the questions; it's their position as a doctor that does that, and it's more in line with overall management*" (nFG2).

**And when I try, it doesn't work.**    Healthcare professionals no longer brought up addictions because of the frustration induced by the failures experienced. Dentists were saying the same thing: *"Because there are lots of little actions like that, you say to yourself, it's like talking to a wall, it's the same thing*!" (dE11) and *"Well, if they don't want to, at any given moment, it's not my fault, that's all (. . .) if people don't want to seize the helping hand, it's too late, it's not our problem (laughs)"* (dE7). Or nurses: *"We try and then after a while, we stop trying*" (nE12*), "They haven't realized the change, well not at all, we can feel that it's not going to work anyway"* (nFG4), *"It's true that sometimes we work a bit in the dark ourselves, eh*?" (nFG1).

## The meeting

**A question of feeling.**    In the approach to addictions, being a paramedic could be an advantage: *"Because often there is a small step to take, I think, and it's true that going to the GP is a bit like going to see your parents, and taking responsibility for what you've done, so you're ashamed of what you've done*" (phE9). The paramedics thought that patients were sometimes afraid of their doctors: *"And when you ask: and you told the doctor?, well no!. He's afraid, he'll get chewed out (laughs)"*. The nurses had the impression of being: "*almost intimate with them, even more sometimes than with the doctor, because we have less of the, you know, the father figure of the doctor*" (phE9). "*And then, we don't have any status, we're a little lower than the doctor in their minds. We're closer to them, we're almost at the same level, we're their physiotherapist, their nurse, their midwife, but we could be their "buddy". It's true, yeah, we go into their homes, we're buddies*" (nFG4).

Ultimately, proximity and personality seemed more important than status: *"I know that the little grannies at the office call me by my first name, we kiss each other, well, sometimes they show up, they're isolated, the family is far away, so you're the grandson, you drop by from time to time. They bring you pancakes, well, you still have a relationship that's much more intimate, much more personal*" (ptFG4). *"I think that between two people, there are things that happen, and things that don't happen*" (nE12).

**The 'bonus' of each profession.**    Besides, each profession has its advantages. It was about physical contact for the physiotherapists: "*it's true that in physiotherapy, we are calm with the patients, we are close, we are tactile, especially in the office*" (ptE13), "*the fact of touching, you enter into the intimacy of the other, and very often at some point, people end up confiding*" (ptFG3). For midwives, it was the habit of addressing intimate issues: "*We become intimate with each other. Because we ask them how their sexuality is going, so in the end addictions are less intimate*" (mwFG3). And the opportunity to act during a privileged time frame: "*There are people for whom the time of pregnancy is a sufficiently powerful motivation, to decrease or stop. Besides, it's a discourse they hear well at this point in their lives*" (mwE6). The nurses had the advantage of receiving confidences: "*I hear a lot of people say that there's not enough listening at the GP level so they say. Well, at least with you we can talk*" (nFG2). The 'bonus' for pharmacists was their knowledge of medications: "*We really have warnings, we know that a certain medication is more likely to cause addiction, we can really spot this kind of thing*" (phE1). Midwives, physiotherapists, and pharmacists, also highlighted their availability: "*During pregnancy follow-up, you see them regularly, so you can discuss it more easily*" (mwE4), "*Patients have more time to ask questions at the pharmacy*" (phE2) and "*What is interesting in the physiotherapist's job, is that we have time with patients repeatedly*" (ptE10).

**Individual management sometimes far from care.**   When meetings took place between professionals and patients, a certain inertia could eventually result, where respect for liberties seemed to prevail over the prevention of an emerging disorder: "*Adults who are adults and vaccinated, and who smoke a joint or two a month, that's their problem, honestly, uh, it's like someone who's going to drink a little bit too much, who gets a little drunk during the month, well, as long as he doesn't drive, let's say, it's his liver, it's his organs, he does what he wants (Laughs)*" (dE7), or "*Everyone is free to do what they want! If it only puts their life in danger, it's no problem*" (nE3).

The meaning of certain comments showed a singular approach. Pharmacists sometimes adopted a commercial discourse as "*clients on methadone. . .*" (phE5). Nurses and dentists appeared to be looking for confessions, rather than confidences: "*We tried to reach out to him a little bit to get him to tell us the truth*" (nFG4), "*depending on how much alcohol the patient confessed*" (dE7). Sharing the identification of a substance use disorder, among the healthcare team, was not systematic, or criticized as denunciation: "*We're not cops, so. . . people do what they want*" (nE3), "*it's not in my values to denounce people, so. . .*" (nE3), "*We're not here to be the police. (Laughs)*" (nFG1). Some people seemed to confuse betrayal with doctor-patient confidentiality: "*It would really feel like betrayal to me, if we called the doctor behind the patient's back*" (nFG4).

## Debriefing

There was, however, a desire for coordination and teamwork: "*We often tell them that we are a team in front of them. Yes, we talk about your health as a team, so what you tell me, okay, it's professional secrecy, but if we consider that the doctor needs to know, we'll let him know*" (nFG4). "*We're all here to participate in the same thing, because if we detect them, advise them, or direct them, at some point, they're going to pass into the hands of the doctors*" (nFG2). Insufficient screening for addictions in primary care is reflected in the fact that participants had less to say about debriefing.

## Discussion

This study explored the practice and experience of screening addictive disorders, by primary care paramedics. A specific observation post was found for each paramedical professional. The identification of an addictive disorder requires taking the time to observe. Being a local healthcare provider allows immersion in people's real lives, and home visits give them a privileged position, for observing risk behaviours [26,27]. In this respect, primary care paramedics recognize that they have a role to play in identifying addictions, which is complementary to that of doctors. These strengths, along with profession-driven competencies in screening practice, have already been described in a review of literature, on nurses and social workers [28]. However, there is ambivalence in their discourse, since they feel it is not their role to address the subject. Thus, they sometimes remain silent witnesses. If they consider that the doctor can ask questions that are not related to the reason for the consultation, primary care paramedics feel the duty to remain within the framework of the care they are providing. They do not allow themselves to do what they think is the responsibility of the general practitioner. Paramedics' reluctance to broaden their scope of practice in this field was also described [29]. These representations must be taken into account, to enable the delegation of tasks to be developed concerning technical procedures, but also the patient-centric care desired by the reform of the healthcare system [30]. To enhance the ability and willingness to engage therapeutically with patients with addictive disorders, adequate undergraduate courses and promotion of interprofessional models that optimize the strengths of each profession, are already desired [28,29]. In this manner, screening, as recommended by the SBIRT strategies, cannot be limited to

unilateral screening, already conceptualized in the discourse of addictologists, where only the healthcare provider notes the disorder [31–33]. A "shared screening" would allow us to engage in dynamic care [34].

The reluctance of caregivers expressed here echoes the concept of self-censorship already found in the study exploring the point of view of addictologists [33]. The fear of rushing patients, of making them feel guilty, of not knowing how to react, of running out of time, of breaking the healthcare provider-patient bond were recalled by the paramedical professionals. Exploring their discourse directly brings a new origin to this self-censorship: paramedics do not feel as legitimate as general practitioners in dealing with these intimate subjects. A feeling of inaudibility towards the patient is also expressed. This limit, perceived as a transgression, has already been described in fields other than addict [35]. Thus, everyone keeps a practice circumscribed to their usual acts and develops an organo-centric vision, like technicians: care prescribed by the doctor for nurses and physiotherapists, pregnancy for midwives, prescription for pharmacists, and the oral sphere for dentists. This prism of restrictive activity induces a professional scotoma responsible for a trivialization: not asking the question of an addiction reinforces the disorder. For the patient, the caregiver's omission means that he or she is condoning consumption [36–40].

The time at which the addiction is confided by the patient to the caregiver seems to be influenced by several factors. According to professionals, the relationship established with the patient appears more conditioned by personality, than by status. It was recalled that a listening, caring, equal and non-stigmatizing relationship is necessary, which had also been found when exploring the point of view of addicted patients [19]. Proximity identification with an empathic and patient-centric approach would be more effective, than a systematic approach, the main thing being to move away from a possible social identity [40]. Paramedics also consider that they are more affordable, and would have more time to devote to the patient, to develop this relationship conducive to their disclosure. A buddy relationship between caregiver and patient would encourage disclosure. A recent qualitative study that asked patients about the ideal conditions for disclosure, also reported that patients preferred to confide in a trusted caregiver, rather than a specialist seen on an *ad hoc* basis [41]. This facilitating proximity in the patient's disclosure expressed, here is in contrast to what addictologists described as a relational routine not very conductive to disclosure: the more intimate one would become, the less one would dare to ask intimate questions, despite the climate of trust [33]. It is not surprising that the opinions of primary and secondary care professionals diverge, and it is reassuring that a long-term follow-up relationship does not preclude comprehensive patient management.

The management of an addictive disorder by paramedics sometimes appears to be far from the recommendations: when confessing to an addiction, but there does not seem to be any immediate bio-psycho-social repercussions according to the paramedics, these latter seem to minimize and trivialize this consumption disorder. While the proximity relationship facilitates disclosure, it would also appear to induce inertia in their care and in the sharing of information within the medical team. The paramedics' discourse on addictology includes all the elements that lead to therapeutic inertia, such as lack of training and motivation, as in the case of other chronic conditions monitored in primary care, such as hypertension [42]. A recent study has shown that an interprofessional SBIRT training program, can provide gains in terms of knowledge, confidence, and skills, and thus reduce the delay in screening [43].

### Strengths and weaknesses

Qualitative research has some weaknesses and specificities. We tried to limit investigation bias by using open questions and an open frame. We also attempted to limit the verbatim

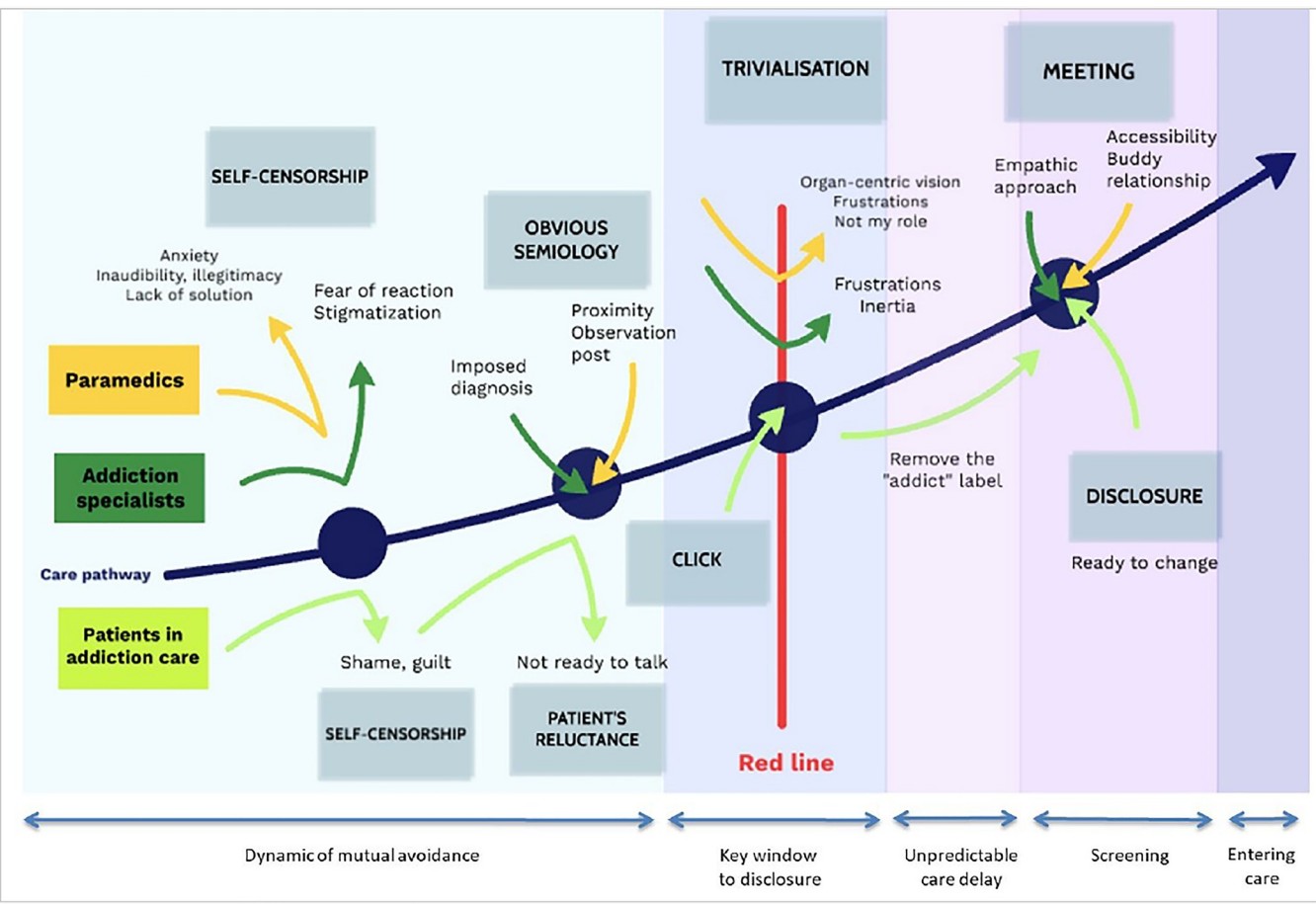

**Fig 1. Conceptualizing caregivers' practice and experience of screening for addictive disorders.**

interpretation bias by using independent coding, by two researchers blinded to each other's decision, and data triangulation. Unexpected comments showed that the exchanges were free. These focus groups seemed appropriate for nurses, physiotherapists, and midwives, whose practices share certain commonalities, such as scheduled consultations and home visits. They could thus, easily exchange their practice, even though the professions differed. The flexibility of the individual interviews also allows for more personalized follow-up, to confirm or deepen certain elements, that emerged during the focus groups. This mixed format was well-accepted by participants. Participants working in the authors' close social network may have induced a social desirability bias during the interviews or focus groups. Ideally, participants unknown to the authors should be recruited.

Using a grounded analysis theory allowed us to access the details of the different points of view, to explicit the "how", and not only the "why" as the usual "insufficiently trained" described by thematic analysis in a recent study [29]. The choice of grounded theory analysis was justified by the search for the conceptualization of tracking, by primary care allied health professionals. Based on previous qualitative studies on the theme of screening for addictions in primary care [33,40], a graphic representation of the concepts that emerged is presented in Fig 1.

In essence, the type of exploration conducted here gathers subjective data. The same applies to their analysis. To ensure the validity of this study, the scientific criteria of the qualitative methods of the COREQ grid were respected (S2 Appendix).

## Conclusions

This qualitative study underlines the inadequacy between the specialization of healthcare professionals, through increasingly technical job references, and the current willingness of public authorities, to encourage multidisciplinary care that mobilizes skills such as the patient-centric approach. The nature of the relationship between professionals and patients remains to be explored since an almost "friendly" relationship seems to favour disclosure, but also to induce inertia in care. This study complements the studies already conducted with addictologists and addict patients. These results will have to be compared with the opinions of general practitioners and patients received in primary care, for a more global model.

## Supporting information

**S1 Appendix. Initial interview guide.**
(DOCX)

**S2 Appendix. COREQ (COnsolidated criteria for REporting Qualitative research) checklist.**
(DOCX)

## Acknowledgments

We would like to thank research and innovation department, Hospital of Tours. This article is supported by the French network of University Hospitals HUGO ('Hôpitaux Universitaires du Grand Ouest').

## Author Contributions

**Conceptualization:** Jean Pierre Lebeau, Maxime Pautrat.

**Data curation:** Agathe Edeline.

**Formal analysis:** Agathe Edeline, Maxime Pautrat.

**Investigation:** Agathe Edeline, Amelie Tripault.

**Methodology:** Jean Pierre Lebeau, Maxime Pautrat.

**Supervision:** Jean Pierre Lebeau, Maxime Pautrat.

**Validation:** Jean Pierre Lebeau, Maxime Pautrat.

**Writing – original draft:** Agathe Edeline, Maxime Pautrat.

**Writing – review & editing:** Agathe Edeline, Amelie Tripault, Jean Pierre Lebeau, Maxime Pautrat.

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
