## [Decision Letter · Decision Letter 0]

8 Oct 2024

PONE-D-24-11025Cross-analysing the opinions and experiences of nurses, physiotherapists, dentists, midwives and pharmacists with respect to addictive disorder screening in primary care: a qualitative studyPLOS ONE

Dear Dr. pautra,

Thank you for submitting your manuscript to PLOS ONE. After careful consideration, we feel that it has merit but does not fully meet PLOS ONE’s publication criteria as it currently stands. Therefore, we invite you to submit a revised version of the manuscript that addresses the points raised during the review process.

We look forward to receiving your revised manuscript.

Kind regards,

Mohammad Sidiq, PhD Pain Sciences Physiotherapy

Academic Editor

PLOS ONE

Journal Requirements:

4. Please remove your figures from within your manuscript file, leaving only the individual TIFF/EPS image files, uploaded separately. These will be automatically included in the reviewers’ PDF. 

**Additional Editor Comments:**

Dear Authors kindly find my suggestions and Reviewer comments:

1. Elaborate a better reason for using grounded theory as the chosen qualitative approach and contemplate indicating how it excels against some other similar strategies.

2. Increase the number of specific ideas for an appropriate approach to training or policy change that may improve allied health worker involvement in addiction screening.

3. Use tabular form or an organizational chart to recapitulate the potential findings as well as the enablers and challenges related to the HCPC in various workers. This will help the readers understand the complexity of the students’ relationships that was detected in the study, at first glance.

4. The results can be compared with the international literature on the addiction screening so the study is more generalizable and can be applied for the other countries besides French healthcare context.

5. Consider several aspects of sample heterogeneity – geographic and professional – particularly in terms of the contrast between urban and rural practices for the improvement of your samples’ representativeness.

Reviewers' comments:

Reviewer's Responses to Questions

**Comments to the Author**

1. Is the manuscript technically sound, and do the data support the conclusions?

Reviewer #1: Partly

Reviewer #2: Yes

2. Has the statistical analysis been performed appropriately and rigorously? 

Reviewer #1: N/A

Reviewer #2: Yes

3. Have the authors made all data underlying the findings in their manuscript fully available?

Reviewer #1: No

Reviewer #2: Yes

4. Is the manuscript presented in an intelligible fashion and written in standard English?

Reviewer #1: Yes

Reviewer #2: Yes

5. Review Comments to the Author

Reviewer #1: Thank you for the opportunity to review this article. I found it to be an interesting read and believe that there is not a lot of research done in this important area. However, I would suggest the following minor and major edits to improve the piece:

Data availability -

It is recommended that authors include selected verbatim interview quotations and focus group data underlying the findings.

Abstract -

This is well written and gives a nice overview of the paper!

Consider the following minor points for improvement:

1. Consider adding another 1-2 keywords.

2. The spelling of the word ´analysing´ is spelt differently in the full and short title. Authors should ensure this is spelt similarly throughout the paper.

3. Consider combining the background and objective section of the abstract. Consider adding the main aim(s) of the research.

4. Correct the spelling error of the word between in the methods section of the abstract:

“Methods: This qualitative study inspired by the grounded theory was carried out betwenn August 2018 and July 2019.”

Introduction -

Consider the following major and minor points for improvement:

The introduction is well written. However, I found it very short and lacking relevant information. There are also several grammatical, spelling, and punctuation errors throughout which I would advise authors to correct as well as the rest of the paper.

5. Punctuation is missing throughout the introduction. In the first sentence of the first paragraph of the introduction, add a comma after the word mortality, and in the second sentence, add a comma after the word opioids. Second paragraph of the introduction, add a comma after the word inefficiency and after the word midwives in both sentence 3 and sentence 7. Correct the spelling of the word `midwifes´. The authors should check for further missing punctuation, spelling and grammar errors throughout the manuscript.

“The global burden of addiction disorders is based on their morbidity, mortality and social costs1,2. Alcohol, opioids and cannabis are the most prevalent, with a risk factor for premature death and disability3–5.”

“But many already known obstacles to addictive disorder screening in primary care remain, such as lack of time, a feeling of inefficiency and patient reluctance14–19.”

“Since 2016, dental surgeons, nurses, midwifes and physiotherapists have been called upon to carry out this screening21(p201).”

“Since 2016, dental surgeons, nurses, midwifes and physiotherapists have been called upon to carry out this screening21(p201).”

6. Authors could give examples or add to the sentence in the first paragraph of the introduction to make readers more familiar with the DSM-V criteria:

“The fifth edition of the DSM-V described the criteria for diagnosing dependence for both drug related disorders and non-drug-related behaviours7.”

7. Authors should extend/give a brief explanation to readers as to what the Screening Brief Intervention and Referral Treatment is upon first mentioning in the introduction.

8. The authors should consider adding to their main research questions after mentioning the aim of the study at the end of the introduction.

9. Authors should also consider adding why the study was developed, why it is important, and what benefits will it have/add to research.

Methods -

Consider the following major and minor points for improvement:

10. Punctuation missing in the participant’s section.

11. Authors should include how the participants were recruited and via what means?

12. Authors should make the inclusion criteria clearer in the participant's section as well as add how many participants were included in the study.

13. What is the icebreaker question mentioned in the data collection section? Authors should add this and/or include examples.

14. Consider adding additional sections here to further explain the methods (it may be useful to add a semi-structured interview section, as well as a focus group section and explain what they are, how they were used in the study, etc.).

15. Authors should add a sentence or two to explain to readers what the grounded theorization approach mentioned is in the analysis section.

16. Authors should check for any grammatical errors, spelling, and punctuation throughout the methods section.

Results -

Consider the following major and minor points for improvement:

17. Authors switch between fully written out numbers and numerals. Check consistency throughout the paper.

18. The authors should consider including a table of the themes and subthemes to the results section to make these clearer for the reader.

19. Consider renaming some of the theme and subtheme headings to be more technically sound/scientific and less vague

20. Authors could make it clearer as to what information in the results section is from the semi-structured interviews and what is from the focus groups.

21. The authors should connect each theme back to the research questions and in the discussion.

22. Subtheme titled `The “bonus” of each profession`- authors should distinguish between using double and single quotation marks when directly quoting participant interviews and headings. Consider using single quotations in this title and in text (i.e., The ´bonus´ of each profession). Check for consistency throughout paper.

23. Consider adding to the Debriefing theme. This appears very short compared to the others.

24. Authors should check for any grammatical errors, spelling, and punctuation throughout the results section.

Discussion -

Consider the following major and minor points for improvement:

25. It is recommended that authors provide more information when linking their findings back to past research. Authors should consider doing this throughout the discussion section.

26. Authors should distinguish between using double and single quotation marks when directly quoting participant interviews and headings.

27. Authors should avoid directly quoting participants in their discussion section – this is for the results section. It is recommended that the discussion should be used merely to discuss the findings and link back to past studies.

28. Authors should check for any grammatical errors, spelling, and punctuation throughout the discussion section.

29. Authors should consider re-writing and keeping the strengths and weaknesses section short with a specific focus on the strengths, limitations, and future directions of the study (1-2 paragraphs). Authors include a lot of information that should be in the methods section of the paper. The authors should consider moving relevant method information earlier (i.e., how focus groups were arranged, took place, etc.), and remove repetition and other irrelevant information.

30. Figure 1 is blurry.

Conclusion -

Conclusion is to the point and easy to read!

Consider the following minor point for improvement:

31. The first sentence of the conclusion is identical to the first sentence of the discussion. Authors should consider re-writing to avoid repetition.

Reviewer #2: The complexity of primary care paramedics' viewpoints on addiction screening is aptly captured by the qualitative approach.

Additional comments: Give a more thorough explanation of how the recruiting process might affect the results and make recommendations for how to deal with this in subsequent studies.

6. PLOS authors have the option to publish the peer review history of their article (what does this mean?). If published, this will include your full peer review and any attached files.

Reviewer #1: No

Reviewer #2: No

---

## [Author Response · Author response to Decision Letter 0]

28 Nov 2024

Response to editor and reviewers

Additional Editor Comments:

Dear Authors kindly find my suggestions and Reviewer comments:

1. Elaborate a better reason for using grounded theory as the chosen qualitative approach and contemplate indicating how it excels against some other similar strategies.

Response = We added this main reason in the discussion section : “The choice of grounded theory analysis was justified by the search for the conceptualisation of tracking by primary care allied health professionals”.

2. Increase the number of specific ideas for an appropriate approach to training or policy change that may improve allied health worker involvement in addiction screening.

Response = We have expanded the bullet points so that readers can easily find the key messages for their future practice.

3. Use tabular form or an organizational chart to recapitulate the potential findings as well as the enablers and challenges related to the HCPC in various workers. This will help the readers understand the complexity of the students’ relationships that was detected in the study, at first glance.

Response = We are sorry, but we're not sure we understand what you're waiting for.

4. The results can be compared with the international literature on the addiction screening so the study is more generalizable and can be applied for the other countries besides French healthcare context.

Response = You're right, and that's how we present our results in a global model based on data found in the literature, with no specific French healthcare system.

5. Consider several aspects of sample heterogeneity – geographic and professional – particularly in terms of the contrast between urban and rural practices for the improvement of your samples’ representativeness.

Response = Our sample reflects the diversity of professions and types of practice most common among paramedics, as show tables 1 and 2. 

Comments to the Author

Reviewer #1: Thank you for the opportunity to review this article. I found it to be an interesting read and believe that there is not a lot of research done in this important area. However, I would suggest the following minor and major edits to improve the piece:

Data availability -

It is recommended that authors include selected verbatim interview quotations and focus group data underlying the findings.

Response = Thank you for your recommendation. From our perspective, the selected verbatim quotations included in results section illustrate and support our findings. We have carefully chosen these excerpts to highlight the most relevant and representative aspects of the participants' perspectives. Adding more quotations might risk diluting the key messages or overloading the reader with redundant information.

Abstract -

This is well written and gives a nice overview of the paper!

Consider the following minor points for improvement:

1. Consider adding another 1-2 keywords.

Response = We added this three: addictive disorders, cross-analyzing, early detection

2. The spelling of the word ´analysing´ is spelt differently in the full and short title. Authors should ensure this is spelt similarly throughout the paper.

Response = Thank you, we preferred analyzing and we changed it. 

3. Consider combining the background and objective section of the abstract. Consider adding the main aim(s) of the research.

Response = We have combined the background and objective sections. 

4. Correct the spelling error of the word between in the methods section of the abstract:

“Methods: This qualitative study inspired by the grounded theory was carried out betwenn August 2018 and July 2019.”

Response = Thank you, this has been changed.

Introduction -

Consider the following major and minor points for improvement:

The introduction is well written. However, I found it very short and lacking relevant information. There are also several grammatical, spelling, and punctuation errors throughout which I would advise authors to correct as well as the rest of the paper.

5. Punctuation is missing throughout the introduction. In the first sentence of the first paragraph of the introduction, add a comma after the word mortality, and in the second sentence, add a comma after the word opioids. Second paragraph of the introduction, add a comma after the word inefficiency and after the word midwives in both sentence 3 and sentence 7. Correct the spelling of the word `midwifes´. The authors should check for further missing punctuation, spelling and grammar errors throughout the manuscript.

“The global burden of addiction disorders is based on their morbidity, mortality and social costs1,2. Alcohol, opioids and cannabis are the most prevalent, with a risk factor for premature death and disability3–5.”

“But many already known obstacles to addictive disorder screening in primary care remain, such as lack of time, a feeling of inefficiency and patient reluctance14–19.”

“Since 2016, dental surgeons, nurses, midwifes and physiotherapists have been called upon to carry out this screening21(p201).”

“Since 2016, dental surgeons, nurses, midwifes and physiotherapists have been called upon to carry out this screening21(p201).”

Response = Thank you, we checked and did the corrections. 

6. Authors could give examples or add to the sentence in the first paragraph of the introduction to make readers more familiar with the DSM-V criteria:

“The fifth edition of the DSM-V described the criteria for diagnosing dependence for both drug related disorders and non-drug-related behaviours7.”

Response = You’re right, it could be clearer. We added this: 

“These criteria comes together four : loss of control, physical dependence, social problems, and risky consumption”.

7. Authors should extend/give a brief explanation to readers as to what the Screening Brief Intervention and Referral Treatment is upon first mentioning in the introduction.

Response = We clarified the SBIRT concept, in introduction section: “SBIRT is a prevention tool for professionals, to identify risky substance use among the patients, with a view to reducing it”.

8. The authors should consider adding to their main research questions after mentioning the aim of the study at the end of the introduction.

Response = Thank you for your comment on this. This has been added at the end of the introduction section.

9. Authors should also consider adding why the study was developed, why it is important, and what benefits will it have/add to research.

Response = We added this sentence. We hope that these changes will highlight the novel contribution that this study offers: “Understanding what healthcare professionals think about the screening of addictive disorders in their day-to-day practice could help to identify some unknown barriers in their appropriation of the SBIRT protocol. As a first step, it would be useful to explore their ability to screen their patients and, possibly, develop a relevant intervention to encourage them to discuss addiction with their patients and improve early screening”. 

Methods -

Consider the following major and minor points for improvement:

10. Punctuation missing in the participant’s section.

Response = We apologize for the oversight. The necessary punctuation has been added in the participants' section.

11. Authors should include how the participants were recruited and via what means?

12. Authors should make the inclusion criteria clearer in the participant's section as well as add how many participants were included in the study.

13. What is the icebreaker question mentioned in the data collection section? Authors should add this and/or include examples.

14. Consider adding additional sections here to further explain the methods (it may be useful to add a semi-structured interview section, as well as a focus group section and explain what they are, how they were used in the study, etc.).

15. Authors should add a sentence or two to explain to readers what the grounded theorization approach mentioned is in the analysis section.

Response = Thank you for your suggestions in remarks from 11 to 15. We have taken the liberty of addressing them together. Indeed, we have completely revised the participants and data collection sections of the method section. Upon reflection, we agree that we have not put enough emphasis on the method of this study. We propose the following rewrite : 

METHODS

This qualitative study recruited healthcare professionals between August 2018 and July 2019. Using a grounded theory approach, enabled investigators to build a model of healthcare professional’s perspectives on addictive disorder screening in primary care. 

Participants

Healthcare professionals included were physiotherapists, nurses, midwives, pharmacists, and dentists. They were recruited from primary care practices in the Centre Val de Loire, Normandy, and Ile-de France Regions, France. The first professionals have been contacted by phone from the author’s caregiver networks, then others were recruited using a snowball technique. 

We conducted 13 individual semi-structured interviews with healthcare professionals by telephone or at the place of their practice. There were seven women and six men aged between 26 and 60. We also conducted multidisciplinary interviews via focus groups including midwives, nurses and physiotherapists. There were nineteen women and six men aged between 25 and 57. Great variability in terms of sex, age, method and practice characteristics was sought for each professional.

All participants were informed about the study and its objectives and provided informed consent. Only 4 midwives refused to participate, because they did not feel concerned by substance use disorders in their practice.

Data collection

Focus groups have the advantage of encouraging interaction between participants and stimulating inter- and intra-disciplinary exchanges. This method exposes studies to the usual opinion leader and social desirability biases. To limit this desirability bias, along with any external biases, we conducted the focus group in a convivial atmosphere, around a lunch. Individual interviews have the advantage of guaranteeing intimacy, spontaneity and freedom of response during exchanges on a subject charged with representations such as addiction. 

The initial guild interview was developed by all the authors and tested on two volunteer caregivers. It included an icebreaker question, which was “Tell me the story of the last patient with an addiction problem you saw ?”, an invitation to share experiences of successful and unsuccessful patient screening and their role in the identification process. New reminders were added to explore the concepts emerging from the initial analyses. All interviews and focus group were audio recorded and transcribed. All verbatim was coded to anonymize participant identity using ph,E1 for pharmacists's interview for example, or FG1 for the first focus group. A personal logbook collected field notes during the research. At the end of this research, all participants were invited to the presentation of verbatims and results, and some of them came.

16. Authors should check for any grammatical errors, spelling, and punctuation throughout the methods section.

Response = Sorry for that, we did corrections. 

Results -

Consider the following major and minor points for improvement:

17. Authors switch between fully written out numbers and numerals. Check consistency throughout the paper.

Response = Sorry for that, we did corrections. 

18. The authors should consider including a table of the themes and subthemes to the results section to make these clearer for the reader.

Response = Thank you for the suggestion. However, since our analysis is based on Grounded Theory Analysis (GTA), listing themes in a table format is not as straightforward as with generalized inductive analysis. In GTA, themes emerge organically from the data, making hard categorization.

If needed, we are more than happy to provide you a screenshot from our analysis software to show how the themes were derived.

19. Consider renaming some of the theme and subtheme headings to be more technically sound/scientific and less vague

Response = We intentionally chose easy-to-read, memorable headings to enhance reader engagement and practical application.

20. Authors could make it clearer as to what information in the results section is from the semi-structured interviews and what is from the focus groups.

Response = You’re right, it could be not clear although each quotes were cited with notification code FG (Focus Group) or I (interview). To explain more clearly, we added this explanation in the method section : “All verbatim was coded to anonymize participant identity using ph,E1 for pharmacists's interview for example, or FG1 for the first focus group”.

21. The authors should connect each theme back to the research questions and in the discussion.

Response = You're right, the length of a qualitative study can lose the reader among all the verbatim. Figure 1 illustrates the link between the quotes from the results section and the concept from the discussion section. For example: the concept ‘self-censorship’ is recalled by the in vivo words ‘ lack of solution’.

22. Subtheme titled `The “bonus” of each profession`- authors should distinguish between using double and single quotation marks when directly quoting participant interviews and headings. Consider using single quotations in this title and in text (i.e., The ´bonus´ of each profession). Check for consistency throughout paper.

Response = This has been changed.

23. Consider adding to the Debriefing theme. This appears very short compared to the others.

Response =Thank you for your comment. The debriefing theme is indeed shorter as it reflects a less prominent theme in the data compared to other themes. This has been added into section: Insufficient screening for addictions in primary care is reflected in the fact that participants had less to say about debriefing.

24. Authors should check for any grammatical errors, spelling, and punctuation throughout the results section.

Response = Thank you for pointing that out. We have made the necessary corrections.

Discussion -

Consider the following major and minor points for improvement:

25. It is recommended that authors provide more information when linking their findings back to past research. Authors should consider doing this throughout the discussion section.

Response = We are embarrassed if our results do not appear to be sufficiently related to the literature. An effort has been made, however, as in this paragraph in the discussion section where we link the concept of ‘self-censorship’ by paramedics to the same concept already described by addictologists in ref 33.

26. Authors should distinguish between using double and single quotation marks when directly quoting participant interviews and headings.

Response = This has been changed.

27. Authors should avoid directly quoting participants in their discussion section – this is for the results section. It is recommended that the discussion should be used merely to discuss the findings and link back to past studies.

Response = Those quoting has been removed.

28. Authors should check for any grammatical errors, spelling, and punctuation throughout the discussion section.

Response = Sorry for that, we did corrections. 

29. Authors should consider re-writing and keeping the strengths and weaknesses section short with a specific focus on the strengths, limitations, and future directions of the study (1-2 paragraphs). Authors include a lot of information that should be in the methods section of the paper. The authors should consider moving relevant method information earlier (i.e., how focus groups were arranged, took place, etc.), and remove repetition and other irrelevant information.

Response = Thank you for your valuable comment. The Strengths and Limitations section has been shorten.

30. Figure 1 is blurry.

Response = The resolution of the figure has been revised.

Conclusion -

Conclusion is to the point and easy to read!

Consider the following minor point for improvement:

31. The first sentence of the conclusion is identical to the first

---

## [Editor Report · Decision Letter 1]

4 Dec 2024

Cross-analyzing the opinions and experiences of nurses, physiotherapists, dentists, midwives, and pharmacists with respect to addictive disorder screening in primary care: a qualitative study

PONE-D-24-11025R1

Dear Dr. Pautrat

We’re pleased to inform you that your manuscript has been judged scientifically suitable for publication and will be formally accepted for publication once it meets all outstanding technical requirements.

Kind regards,

Mohammad Sidiq, PhD Pain Sciences Physiotherapy

Academic Editor

PLOS ONE

Additional Editor Comments (optional):

Authors have addressed the issues raised by reviewers, and I am satisfied with the revision, and the manuscript can be accepted for publication. Congratulations to the authors.
---

## [Editor Report · Acceptance letter]

14 Jan 2025

PONE-D-24-11025R1 

PLOS ONE

Dear Dr. Pautrat, 

I'm pleased to inform you that your manuscript has been deemed suitable for publication in PLOS ONE. Congratulations! Your manuscript is now being handed over to our production team.

Kind regards, 

on behalf of

Dr. Mohammad Sidiq 

Academic Editor

PLOS ONE